# Immunotherapy in Urothelial Cancer: Stop When Achieving a Response, Restart upon Disease Progression

**DOI:** 10.3390/cancers15143654

**Published:** 2023-07-18

**Authors:** Youssra Salhi, Ronald De Wit, Debbie Robbrecht

**Affiliations:** Department of Medical Oncology, Erasmus MC Cancer Institute, Dr. Watermolenplein 40, 3015 GD Rotterdam, The Netherlands; y.salhi@erasmusmc.nl (Y.S.); r.dewit@erasmusmc.nl (R.D.W.)

**Keywords:** urothelial cancer, PD-1, PD-L1, immune checkpoint inhibitors, pembrolizumab

## Abstract

**Simple Summary:**

Immune checkpoint inhibitors (ICI) have dramatically changed the landscape of metastatic urothelial carcinoma (UC). However, optimal duration of treatment with ICI in the metastatic setting remains unclear. On the basis of real-world data from the Netherlands of patients treated with pembrolizumab for metastatic UC, we point out that early discontinuation of ICI seems to be reasonable in a subgroup of patients, especially in patients with responding disease (partial or complete response). It is important to draw attention to this topic and to pursue further efforts to evaluate shorter treatment duration for optimizing patient care and outcomes, but also for keeping healthcare affordable.

**Abstract:**

Background: Since there is no clear consensus on optimal treatment duration of PD-(L)1 targeting checkpoint inhibitors in the setting of urothelial cancer (UC) patients, even patients with durable responses are often treated up to 2 years. It is questionable whether this is necessary and whether quality of life improves when treatment is discontinued earlier and restarted when necessary. Methods: We collected available data from locally advanced or metastatic UC patients within the Netherlands between September 2017 and December 2019 treated with first or second-line pembrolizumab, to evaluate treatment duration, reasons for discontinuation, subsequent treatments and survival outcomes. Results: Data were available from 74 patients: 85% (63/74) of patients had a treatment duration of 12 months or shorter, and in seven out of them, treatment was discontinued for another reason than progressive disease. Two patients (3%) had a treatment duration between 12 and 24 months, and eight patients (11%) completed 24 months of treatment. Survival at data cut-off (1 July 2020) with a median follow-up of 35 months was 100% in patients with partial or complete response (6/7 patients) and treatment duration ≤ 12 months, and 100% in patients treated for 24 months. In total, three patients were re-treated with pembrolizumab upon progressive disease during follow-up. Conclusions: In patients who reach partial or complete response during treatment with a PD-(L)1 targeting checkpoint inhibitor, early discontinuation of treatment with pembrolizumab and restart if necessary seems to be reasonable with preserved favorable outcomes. This article should drive further efforts to optimize the treatment duration for patients who respond to treatment with pembrolizumab.

## 1. Introduction

For around one in four or five patients with locally advanced or metastatic urothelial cancer (UC), a durable response from treatment with first or second-line pembrolizumab is achieved [1,2,3,4,5,6]. Most patients reach this response within the first months of treatment, but in case of good tolerance, treatment is typically continued for 2 years even if there is an ongoing response. This is a consequence of the lack of consensus on the optimal duration of immune checkpoint inhibitors (ICIs) in patients responding to their treatment, and continuation for up to two years or even longer in the phase II and III trials [1,2,3,4,5,6]. However, it is questionable whether treatment can be interrupted earlier in case of responding disease and restarting when necessary. The question has also been raised in other disease areas like melanoma and NSCLC, and has led to initiatives such as the Safe Stop trials (NCT0562673, NTR7502) [7,8,9,10,11].

Moreover, in patients with an objective response who discontinued PD-1 blockade, responses have still been reported after reintroduction of PD-1 blockade at disease progression and observed in patients with either stable or partial/complete response at first best response to previous ICI treatment [12,13].

Treatment with ICIs is associated with a broad spectrum of potential toxicities, which can develop rapidly and may persist lifelong, impacting a patients’ health-related quality of life [14]. Also, the incidence of toxicity increases with treatment duration, and in addition, treatment with ICI requires regular visits at the hospital and has a large financial burden on society. Therefore, if early discontinuation of PD-(L)1 targeting ICI would be feasible in UC patients, this would bring several advantages.

Although robust evidence on early discontinuation of PD-(L)1 targeting ICI and the support of biomarkers in this context is lacking, currently many treating physicians do discuss early discontinuation with their patients on an individual basis. Therefore, we evaluate the available data from patients within the Netherlands treated with pembrolizumab for UC, to look for patterns concerning treatment duration, reasons for discontinuation, subsequent treatments and survival outcomes to reduce toxicity and costs. We also evaluated differences in clinical parameters, and if available, PD-L1 status and circulating tumor DNA (ctDNA)-based measures.

## 2. Materials and Methods

Data were available and collected from 74 patients with locally advanced or metastatic UC who were treated with pembrolizumab in a three-week schedule (200 mg q3w) between 1 September 2017 and 21 December 2019 in the Netherlands. Data were collected in compliance with the Declaration of Helsinki and the International Conference on Harmonization Good Clinical Practices Guidelines, and applicable informed consent documents were reviewed and approved by the regional review boards and ethics committees. Maximum duration of treatment was 24 months and reasons for earlier discontinuation of therapy were progressive disease (PD), pembrolizumab-related toxicity or a drug holiday in a shared decision setting between treating physician and patient. Prior to the start of therapy and until PD, tumor response was evaluated according to response evaluation criteria in solid tumors v.1.1 and defined as stable disease (SD), partial/complete response (PR/CR) or PD. The PD-L1 status on tumor tissue of every patient was performed by the Dako PD-L1 IHC 22C3 assay. PD-L1 protein expression in urothelial carcinoma was determined by using the Combined Positive Score (CPS), which is the number of PD-L1 staining cells (tumor cells, lymphocytes, macrophages) divided by the total number of viable tumor cells, multiplied by 100. The specimen should be considered to have PD-L1 expression if CPS ≥ 10.

## 3. Results

Overall, 26 out of 74 patients (35%) had at least stable disease and ongoing treatment six months after initiation of pembrolizumab, and 48 patients (65%) developed progressive disease during treatment.

In total, 63 out of 74 patients (85%) had a treatment duration of 12 months or shorter. In seven out of these 63 patients, treatment was discontinued for another reason than progressive disease; Pembrolizumab related toxicity in four patients (diarrhea *n* = 2, arthritis/arthralgia *n* = 2) and three patients it was decided to introduce a drug holiday because of ongoing disease response. Eight patients (11%) completed 24 months (35 administrations of pembrolizumab), 11 patients (15%) had a treatment duration between 12 and 24 months, and two out of 11 patients had a wish to pause treatment with pembrolizumab in the absence of progressive disease.

### 3.1. Patients with Treatment Duration ≤ 12 Months

These seven patients had a median treatment duration of 8.5 months, and disease status at the moment of pembrolizumab discontinuation was either SD (*n* = 1), PR (*n* = 3) or CR (*n* = 3). The characteristics of these patients are summarized in Table 1 and Table 2. Five out of seven patients had PD-L1-positive disease (combined positive score ≥ 10), and no patients had visceral metastases. At the time of data cut-off (1 July 2020), the median follow-up period was 35 months and five patients (71%) were still alive (Figure 1). One patient was re-treated with pembrolizumab upon progressive disease with ongoing stable disease at data cut-off. One patient was re-treated with stereotactic radiotherapy of the primary tumor followed by pembrolizumab, reaching complete response for the second time. The other two patients were treated with docetaxel (only one cycle) and carboplatin/gemcitabine, respectively, as the next line of treatment. Figure 2 illustrates two out of seven patients treated <12 months and a durable response of disease.

### 3.2. Patients with Treatment Duration > 12 to <24 Months

Two patients had a median treatment duration of 14.6 months with a PR at the moment of pembrolizumab discontinuation, and both patients had PD-L1-positive disease and visceral metastases (Table 1 and Table 2). Data on subsequent therapy after pembrolizumab and survival are missing; therefore, these patients are not included in Figure 1.

### 3.3. Patients with Treatment Duration of 24 Months

Eight patients completed 24 months of treatment with pembrolizumab. The characteristics of these patients are reported in Table 1 and Table 2. Disease status in these patients at the moment of discontinuation was either PR (*n* = 5) or CR (*n* = 3). Six of eight patients had PD-L1-positive disease, and four of eight patients had visceral metastases. One patient started with re-treatment of pembrolizumab, and one patient underwent chemoradiation of the bladder with curative intent. The five patients who did not develop disease progression had ongoing response and no subsequent treatment, and all patients were alive at data cut-off (1 July 2020) with a median follow-up of 35 months (Figure 1).

### 3.4. ctDNA-Based Measures

In all patients, data were available from the ctDNA-based Fast Aneuploidy Screening Test-Sequencing System (mFast-SeqS), a low coverage sequencing approach to detect aneuploidy in circulating free DNA, which is a correlate of ctDNA levels [15]. A high aneuploidy score (≥5) in UC patients has been shown to be associated to non-response to pembrolizumab and worse outcomes [16]. A major advantage of using this test prior to treatment with pembrolizumab, at baseline, is its ability to support the decision whether or not to initiate pembrolizumab. But it might also be helpful on the decision whether or not a patient might be a candidate for early discontinuation of treatment, both at baseline and during the course of the treatment.

In the seven patients with treatment duration ≤ 12 months, there were no patients with a high mFast-SeqS aneuploidy score at baseline. Also, all three patients who were treated 12 to 24 months had a low aneuploidy score. In the patients with 24 months of treatment, only one patient had a high aneuploidy score.

## 4. Discussion

Early discontinuation of PD-(L)1 targeting ICI in UC patients might be feasible in a subset of patients. Since early discontinuation of this class of expensive drug has the potential to majorly positively impact patient and healthcare-related aspects and costs, this has been fuel to discussion among treating physicians. However, there is no standard approach or clear consensus to treatment discontinuation in responding patients. Whether treatment can be safely discontinued in responding patients or should be continued remains controversial due to several limitations in current studies, such as variability in study designs, highly selective eligibility criteria hampering representation of real-world patient populations and inadequate documentation regarding treatment discontinuation.

When evaluation UC patient data from September 2020 till December 2019, a time period in which pembrolizumab had become the standard of care in the second-line setting and for PD-L1-positive disease cisplatinum-unfit patients in the first-line setting, we observed that indeed there seems to be a subgroup of UC patients in which early discontinuation of pembrolizumab seems to be justifiable.

Several studies in different types of solid tumors (e.g., melanoma and non-small cell lung cancer) have found that a shorter duration of treatment may be sufficient. Studies investigating the optimal duration of immunotherapy have in common that authors conclude that discontinuation of ICI treatment seems to be possible in a subgroup of patients [8,10,17,18]. However, an explorative analysis of the Checkmate-153 in NSCLC suggested that outcomes are superior when ICI therapy is continued beyond one year instead of treating for a fixed period of one year [19]. In addition, when deciding to terminate treatment prematurely, efficacy of re-treatment in case of disease progression should be maintained. The literature regarding re-treatment with anti-PD-(L)1 treatment is scarce. However, several small cohorts in advanced melanoma reported data regarding the efficacy of re-treatment. In the KEYNOTE-006, in which 15 patients were treated with a second course of pembrolizumab, seven patients achieved renewed tumor response [20]. The cohort described by Jansen et al. reported that six out of 19 patients who received subsequent therapy with an anti-PD-(L)1 inhibitor responded to therapy [21]. Other studies included a small group of patients [22,23,24,25,26,27].

In the landmark trials in advanced or metastatic UC, patients were treated with pembrolizumab for a duration of 2 years. Only in a minority of patients was treatment discontinued earlier for a reason different from progressive disease [28,29]. In the KEYNOTE-045, 26 out of 266 patients completed 2 years of treatment with pembrolizumab, and 10 patients discontinued treatment due to durable complete response before 2 years. In the KEYNOTE-052 study, platinum ineligible patients were treated with pembrolizumab in first-line. A total of 35 out of 106 patients completed 2 years treatment with pembrolizumab, and 35 patients discontinued treatment before progression due to complete response before the end of treatment. These data combined with our real-world data suggest that there is a subgroup of patients where treatment with ICI can be interrupted, and this seems to be reserved for the patients achieving partial or complete response upon treatment with expected durable response after treatment cessation.

How can the right patient be selected for shorter treatment duration? Clinical criteria and biomarkers might be helpful. First, early treatment discontinuation not resulting from toxicity seems to be reserved for patients with a durable disease response and, therefore, a definition of durable response is necessary. In 2019, the definition of durable response was defined as progression-free survival (PFS) of an individual patient exceeding three times the median PFS of the same group [28]. By using this definition, patients with durable stable disease will also be included.

Second, treatment may be discontinued due to immune-related toxicity. In case of grade 1 or grade 2 immune-related toxicity, it is deemed feasible to continue ICI therapy or restart once the toxicity has reduced to grade ≤ 1. However, in case of patients with responding disease, it is debatable whether this is the best strategy. Re-treatment with ICI upon progressive disease has shown to introduce response again, yet not in all patients [12]. Our data showed that patients who relapsed after discontinuing pembrolizumab remained sensitive to pembrolizumab re-challenge. However, the literature underwriting immunotherapy re-challenge in patients with mUC is scarce.

Third, a biomarker to support decision on treatment duration would be very welcome. Tested markers such as PD-L1 expression, tumor mutation burden, somatic mutations and copy number alterations, gene expression profiles and tumor microenvironment factors have in common that implication into clinical practice is challenging due to several challenges like variable predictive values, different cut-offs and assays being used and the requirement of tumor tissue in many cases. Therefore, ctDNA-based measures could serve for promising biomarkers [16,30,31]. Since ctDNA is collected from minimally invasive blood draws and serves as a surrogate for tumor burden, it is an attractive and patient-friendly method to monitor tumor response or progression. ctDNA clearance overtime, or the ctDNA mFast-SeqS aneuploidy score might be potential biomarker candidates, and prognostic and predictive values have been shown recently [16,30,31]. The low resolution approach, without the need to be informed about the genomic makeup of the tumor, makes the mFast-SeqS method an attractive method to use as a support in decisions on treatment discontinuation, as well as treatment re-initiation. In the patient cohort described here, all except one patient had a low mFastSeqS-based aneuploidy score at baseline and prior to cycle 2 pembrolizumab, which is suggestive of the potential to identify a patient population eligible for a shorter course of ICI treatment based on this minimally invasive test.

Fourth, shorter courses of ICI therapy should be evaluated in the context of trials. Randomized trials are not always feasible, but this should not refrain one from studying this important topic. So-called ‘trials within a cohort’ or TwiC designs [32] enable comparing outcomes of patients in existing cohorts in which patients are being treated differently. Single-arm phase 2 trials evaluating a different treatment duration in a patient population matched to the population from the earlier executed randomized phase III trial allow for indirect comparison of data on patient outcomes. Contemporary statistical methods, like the Bayesian method, make it possible to accurately weigh differences between independent cohorts, for example, historical or real-world cohorts and cohorts from randomized trials [33,34].

Lastly, quality of life should be a prioritized aspect in cancer treatment, including metastatic UC patients. Registering quality of life during treatment of these patients seems to be relevant to optimize adherence to treatment and to reduce the frequency of hospitalization due to complications. Recently published studies have shown that the use of patient-reported outcomes (PROs) during treatment may improve better supportive care and symptom monitoring leading to an improvement in clinical outcomes. A survey in 2022 in metastatic cancer patients revealed that the chance of eliminating all the evidence of disease, durability of treatment, improvement of quality of life and the ability to go off therapy are important factors in optimal cancer care [35]. A questionnaire by the KCCure gave some good insights into patients’ perspectives on what is important concerning treatment decisions and the aims of treatments [36]. When it comes to the advice to interrupt a treatment when it is being deemed safe and feasible, 58% of patients were anxious about their cancer progressing, but 80% of patients agreed with discontinuation of treatment. Another 12% of patients felt safer to be able to avoid future side effects. These insights underscore how important it is to really investigate shorter treatment duration in trials in order to inform our patients completely and objectively.

Since January 2022, ICI also have a role in the maintenance setting in UC patients, with avelumab every two weeks in patients with at least stable disease with first-line platinum-based chemotherapy. However, an important hurdle is that within the context of this treatment, avelumab is continued until progressive disease or patient withdrawal based on the JAVELIN bladder 100 trial [37]. Thus, for the maintenance setting, the ‘optimal ICI treatment duration discussion’ is also applicable. The currently ongoing phase 2 Avelumab Short Maintenance trial (EUCTR2020-005781-34-NL) aims to evaluate survival in UC patients who are being treated with maintenance avelumab for a maximum of six months; and within the context of this trial, sequential plasma samples will be collected to look for a potential ctDNA-based biomarker that can support decisions on maintenance initiation and duration of maintenance. Appropriate criteria combined with clinically reliable biomarkers to select patients who will benefit from early discontinuation are needed and should be prospectively examined to maintain efficacy also in the case of early discontinuation.

The described data here have several limitations. They reflect real-world data collected in a time period where DNA sequencing in urothelial cancer patients was not part of standard practice. As a result, information on FGFR status, ERBB2 status, tumor molecular burden, etcetera, are missing. Moreover, the number of patients described in this paper is modest. However, we are convinced that the data presented here are important for the discussion on treatment duration in responding patients.

## 5. Conclusions

In conclusion, shorter courses of PD-(L)1 targeting ICI seems to be feasible in the subgroup of UC patients with responding disease. Tumors that relapse after discontinuation remain sensitive to pembrolizumab re-treatment in most patients, but prospective trials are needed to rule out inferior long-term efficacy and to identify supporting biomarkers. The ctDNA-based modified Fast Aneuploidy Screening Test-Sequencing method holds promise in this respect. Performing trials evaluating shorter courses or less frequent administration schedules is essential for optimizing patient care and outcomes, but also for keeping healthcare affordable.

## Figures and Tables

**Figure 1 cancers-15-03654-f001:**
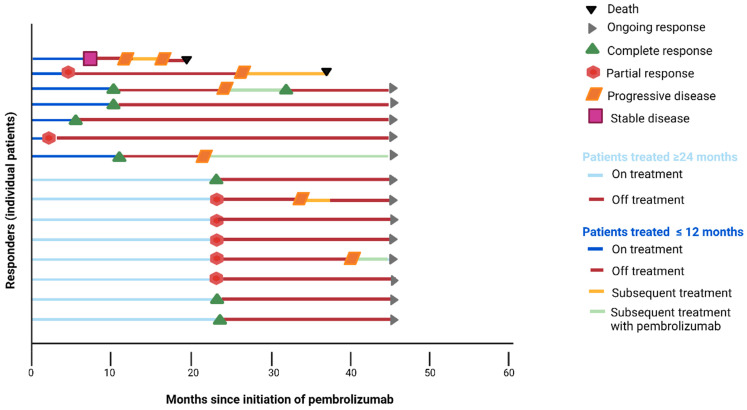
Swimmer plot of patients with advanced or metastatic UC treated with pembrolizumab in second-line setting. The dark blue lines indicate patients who were treated for a maximum duration of 12 months. The light blue lines indicate patients treated 24 months.

**Figure 2 cancers-15-03654-f002:**
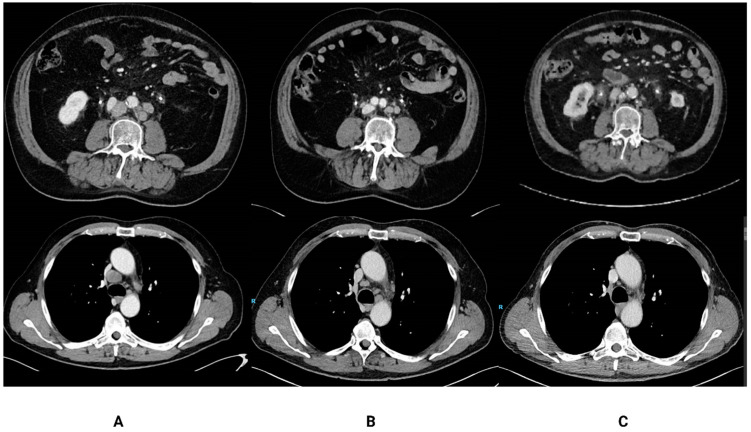
CT-scans of two patients treated <12 months with pembrolizumab at baseline (**A**), just prior to discontinuation of pembrolizumab (**B**) and during follow-up 29 and 41 months resp. after discontinuation of therapy (**C**). The upper row illustrates scans of a patient with non-visceral metastatic disease with discontinuation of treatment with pembrolizumab after eight cycles due to immune-mediated colitis. He developed progressive disease 34 months after discontinuation. The lower row illustrates scans of a patient who also had non-visceral metastatic disease, whose treatment with pembrolizumab was interrupted after seven cycles due to arthritis; this patient has ongoing response up until today (38 months after discontinuation of pembrolizumab).

**Table 1 cancers-15-03654-t001:** Baseline characteristics of patients who were treated ≤12 months, 12 to 24 months or ≥24 months with pembrolizumab.

Characteristic	Treatment Duration ≤ 12 Months (*n* = 7)	Treatment Duration 12–24 Months (*n* = 2)	Treatment Duration ≥ 24 Months (*n* = 8)
**Age (median, years)**	63	71	71
**Gender**			
Male	5 (71%)	2 (100%)	7 (88%)
Female	2 (29%)	0 (0%)	1 (12%)
**Visceral metastases ***			
No	7 (100%)	0 (0%)	4 (50%)
Yes	0 (0%)	2 (100%)	4 (50%)
**WHO-status at baseline**			
0	5 (71%)	0 (0%)	2 (25%)
1	2 (29%)	2 (100%)	6 (75%)
**Line of treatment**			
First line pembrolizumab	3 (43%)	0 (0%)	2 (25%)
Second line pembrolizumab	4 (57%)	2 (100%)	6 (75%)

* visceral metastases were located: liver (*n* = 1), lymph nodes (*n* = 3), lung (*n* = 2), peritoneal (*n* = 1).

**Table 2 cancers-15-03654-t002:** PD-L1 and mFast-SeqS data from patients in accordance to treatment duration. ^ PD-L1 status of the tumor based on the combined positive score (CPS) according to the companion diagnostic PD-L1 staining of pembrolizumab.

Characteristic	Treatment Duration ≤ 12 Months (*n* = 7)	Treatment Duration 12–24 Months (*n* = 2)	Treatment Duration ≥ 24 Months (*n* = 8)
**PD-L1 status ^**			
Negative	2 (29%)	0 (0%)	3 (38%)
Positive	5 (71%)	2 (100%)	5 (62%)
**mFast-SeqS aneuploidy score**			
**at baseline ^#^**			
High (≥5)	0 (0%)	0 (0%)	1 (12%)
Low (<5)	7 (100%)	2 (100%)	7 (88%)
**mFast-SeqS aneuploidy score**			
**prior to cycle 2 pembrolizumab ^#^**			
High (≥5)	0 (0%)	0 (0%)	1 (12%) ^b^
Low (<5)	6 (86%) ^a^	2 (100%)	7 (88%)

**^#^** high aneuploidy score is a score of ≥5, a low aneuploidy score is a score of < 5. ^a^ No mFast-SeqS data available from one patient prior to cycle 2 ^b^ This patient was re-treated with pembrolizumab upon progressive disease.

## Data Availability

Not Applicable.

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
