# Peer review of "Immunotherapy in Urothelial Cancer: Stop When Achieving a Response, Restart upon Disease Progression"

_cancers, 2023, doi:10.3390/cancers15143654_

Round 1

Reviewer 1 Report

The manuscript entitled "Immunotherapy in urothelial cancer: Stop when achieving a response, restart upon disease progression"

The authors, with a collection of some cases, claimed that n patients who reach partial or complete response during treatment with a PD-(L)1 targeting checkpoint inhibitor, early discontinuation of treatment with pembrolizumab and restart if necessary seems to be reasonable with preserved favorable outcomes. 

This work disclosed some interesting points while not solid enough.

Major issues:

1. This work is limited by a small case number.

2. The authors should provide the clinical information of these patients in more detail, the PD-L1 expression status (and clones), tumor differentiaiton, FGFR2/3 mutation status, TMB status should also be provided.

3. The representative images (CT, MRI, and pathology and IHC) should also been provided. 

acceptable

Author Response

Thank you for your thorough review and recommendations. We reply here point-by-point:
1. The work is limited by a small case number. We have added limitations to our discussion to underline this point. 2. The authors should provide the clinical information of these patients in more detail, the PD-L1 expression status (and clones), tumor differentiaiton, FGFR2/3 mutation status, TMB status should also be provided. We have added more information on the used PD-L1 expression test in our methods section. Moreover we have added to our discussion that unfortunately data were missing, predominantly molecular data, since the described patients reflect a real-world patient population and data were collected in a time period where DNA sequencing of tumor tissue from urothelial cancer patients was not part of standard practice. 3. The representative images (CT, MRI, and pathology and IHC) should also been provided. We have now included radiology images from two patients to further illustrate the patient data described in the manuscript. Unfortunately, since the data represent patients treated in 2017-2019 and these patients were treated in different centers in the country, we had to conclude that it was not feasible to receive IHC images before the deadline of re-submission.

Reviewer 2 Report

Dear Authors,

I read with interest the entitled “Immunotherapy in urothelial cancer: Stop when achieving a response, restart upon disease progression” and I congratulate for the effort behind it. 

In this study the authors aimed to evaluate data from 74 patients with locally advanced or metastatic UC treated with pembrolizumab, to look for patterns concerning treatment duration, reasons for discontinuation, subsequent treatments, and survival outcomes; moreover, they sought to explore differences in clinical parameters, and PD-L1 status and circulating tumor DNA based measures.

I found the present study interesting, well written and fluent to read. It concerns with an actual topic, which is of major importance since the lack of consensus on the optimal duration of ICIs in this setting of disease.

The title is descriptive of what authors have explored in their work. The background and scientific rationale for carrying out the study are well presented. Tables and Figures are clear and not repetitive, as well as Results section. Statistical assessment is well conducted and the paper results methodologically correct. Discussion is adequately implemented with the relevant literature, and interpretations and conclusions are well stated and justified by results. I have no concerns or suggestions.

Author Response

Many thanks for your review and judgement. There were no concerns or suggestions to address.

Reviewer 3 Report

The paper deals with the issue of the best protocol to treat mUC patients with ICIs. The paper is well written, easy to follow and provides clinically sound data.

The observation made by the authors and reported in this manuscript is of great relevance since it can provide information to reduce the number of ICI courses during treatment of patients with UC then avoiding unnecessary toxicities. Also relevant is the potential saving for the health system using the proposed therapeutic scheme. 

The use of cfDNA as surrogate of ctDNA, as proposed in this paper, has great potential for routine implementation.

Author Response

(The authors gave the same response as above.)

Round 2

Reviewer 1 Report

The revision is acceptable

Acceptable